# On-Farm Practices Associated with Multi-Drug-Resistant *Escherichia coli* and *Vibrio parahaemolyticus* Derived from Cultured Fish

**DOI:** 10.3390/microorganisms10081520

**Published:** 2022-07-27

**Authors:** Rita Rosmala Dewi, Latiffah Hassan, Hassan Mohammad Daud, Mohd. Fuad Matori, Zunita Zakaria, Nur Indah Ahmad, Saleha A. Aziz, Saleh Mohammed Jajere

**Affiliations:** 1Faculty of Veterinary Medicine, Universiti Putra Malaysia, Serdang 43400, Selangor, Malaysia; ritardewi@yahoo.com (R.R.D.); hassanmd@upm.edu.my (H.M.D.); fuma@upm.edu.my (M.F.M.); zunita@upm.edu.my (Z.Z.); nurindah@upm.edu.my (N.I.A.); saleha@upm.edu.my (S.A.A.); 2Institute of Bioscience, Universiti Putra Malaysia, Serdang 43400, Selangor, Malaysia; 3Faculty of Veterinary Medicine, University of Maiduguri, Maiduguri PMB 1069, Borno State, Nigeria; drmsjajere@unimaid.edu.ng

**Keywords:** aquaculture system, *E. coli*, multi-drug resistance, antimicrobial resistance, *V*. *parahaemolyticus*, risk factor

## Abstract

Aquaculture activities have been implicated as responsible for the emergence of antimicrobial resistance (AMR), leading to broad dissemination and transference of antibiotic resistance to pathogens that affect humans and animals. The current study investigates the on-farm practices and environmental risk factors that can potentially drive the development and emergence of multi-drug-resistant (MDR) *Escherichia coli* and *Vibrio parahaemolyticus* in the aquaculture system. A cross-sectional study was conducted on 19 red hybrid tilapia (*Oreochromis* spp.) and 13 Asian seabass (*Lates calcarifer*, Bloch 1970) farms on the west coast of peninsular Malaysia. Data were collected using a structured questionnaire pertaining to farm demography, on-farm management practices and environmental characteristics. Multi-drug-resistant *E. coli* (*n* = 249) and *V. parahaemolyticus* (*n* = 162) isolates were analyzed using multi-level binary logistic regression to identify important drivers for the occurrence and proliferation of the MDR bacteria. On-farm practices such as manuring the pond (OR = 4.5; 95% CI = 1.21–16.57) were significantly associated with the occurrence of MDR *E. coli*, while earthen ponds (OR = 8.2; 95% CI = 1.47–45.2) and human activity adjacent to the farm (OR = 4.6; 95% CI = 0.75–27.98) were associated with an increased likelihood of MDR *V. parahaemolyticus*. Considering the paucity of information on the drivers of AMR in the aquaculture production in this region, these findings indicate the targeted interventions implementable at aquaculture farms to efficiently abate the risk of MDR amongst bacteria that affect fish that are of public health importance.

## 1. Introduction

Bacteria that are resistant to multiple classes of antibiotics pose increasing challenges in treating animal and human infections. Multi-drug resistance in foodborne pathogens such as *Salmonella* and *Vibrio* are especially of concern because of the prospective challenges of treatment when humans are infected. Foodborne pathogens cause 7.5% (420,000) of all mortalities worldwide [1,2], and approximately 7.7% of the world’s population experience foodborne illnesses annually [3]. In Southeast Asia, 150 million cases of foodborne illnesses are recorded every year, with an annual mortality rate of up to 175,000 [4]. Moreover, Lim et al. [5] estimated 19,000 MDR-related deaths annually in Southeast Asian countries.

The intensification of both terrestrial and aquatic food animal production to improve food sufficiency has been implicated as one of the major drivers of the increasing antimicrobial resistance (AMR). Aquatic ecosystems are vulnerable to various contaminants, including chemicals and drug residues, as they are the recipients of run-offs from multiple terrestrial sources, including agricultural or livestock farms and healthcare facilities [6,7]. Aquaculture activities may increase the antimicrobial resistance burden resulting from the uncontrolled use of antibiotics throughout the production cycle. In fact, certain types of aquaculture products, such as catfish, are linked to antimicrobial use per kilogram that exceeds those in terrestrial animals and humans [8]. Multiple studies have reported significant differences in the levels of antimicrobial resistance genes (ARGs) of bacteria in sediments from an aquaculture setting as compared to non-aquaculture sites [9,10,11,12].

The aquatic environment is a receptacle ideal for bacterial mutation, recombination and horizontal gene transfer [13,14,15]. Increasing AMR levels have been reported among pathogens found in the aquatic environment, such as *Escherichia coli* and *Vibrio parahaemolyticus* [16,17,18]. *E. coli*-associated diarrheal illness is responsible for nearly 200,000 annual deaths globally [2]. Moreover, the multi-drug-resistant commensal *E. coli* strain poses a significant public health risk because *E. coli* is an accomplished AMR genes’ reservoir that is capable of disseminating those genes to other bacteria [19,20]. Meanwhile, more than 20 serovariants of *V. parahaemolyticus*, including 03:K6, 04:K68, 01:K25 and 01:KUT, have been implicated in foodborne disease outbreaks in Asian countries, the United States and several other parts of the world [21].

Aquaculture production has expanded rapidly over the past few years [22] and accounts for more than half of the aquatic food animal production worldwide [23,24]. Southeast Asia provides a large proportion of the aquaculture products, accounting for 22% of world production, and is poised to further expand its production to meet the increasing global demand [25]. Malaysia is one of the Southeast Asian countries where the aquaculture sector is rapidly expanding [26,27]. Unfortunately, the transition of aquatic production to intensification is positively linked to the increasing levels of AMR among various bacteria from the aquaculture settings in this region [28,29,30,31,32,33]. Consequently, the management practices and environmental variations in the aquaculture settings need to be better understood to assign interventions that can most effectively decelerate AMR among bacterial pathogens. In Malaysia, the National Action Plan for AMR includes surveillance of aquaculture products as part of the country’s strategic actions. Our previous paper has discussed the pattern and prevalence of AMR among important bacterial pathogens in aquaculture products and the environment [34]. This study identifies the on-farm management and surrounding environmental factors associated with the detection of MDR *E. coli* and *V. parahaemolyticus* to enable targeted interventions aiming to reduce the occurrence of MDR.

## 2. Materials and Methods

### 2.1. Source of the Isolates

The study design, sampling and collection of data, as well as identification and antimicrobial susceptibility tests, have been described in our previous publication [34]. Briefly, a cross-sectional study was conducted on 32 aquaculture farms in the west coast of peninsular Malaysia. A total of 312 tilapia and 265 Asian seabass fish intestinal contents were sampled. Two hundred and forty-nine (249) *E. coli* isolates were isolated both from fish intestines (*n* = 577) and pond waters (*n* = 32). The isolation and identification of *E. coli* from tilapia and Asian seabass adopted the method of Jang et al. and Ryu et al. [35,36]. Meanwhile, 162 *V. parahaemolyticus* isolates were recovered from the fish intestines (*n* = 265) and pond waters (*n* = 13). The isolation and identification of *V. parahaemolyticus* adopted the method of Huq et al., Momtaz et al., CDC and Neogi et al. [37,38,39,40]. The isolates were tested for antibiotic susceptibility using the disk diffusion method for 12 antibiotics (concentration in µg): cefotaxime (30 µg), ceftiofur (30 µg), ciprofloxacin (5 µg), gentamycin (10 µg), chloramphenicol (30 µg), streptomycin (10 µg), tetracycline (30 µg), ampicillin (10 µg), erythromycin (15 µg), nalidixic acid (30 µg), kanamycin (30 µg) and trimethoprim (5 µg). The susceptibility test against colistin was performed using the broth microdilution method (BMD). These antibiotics were chosen in accordance with the WHO and OIE recommendations for antimicrobial usage in humans and livestock [41,42], as well as Malaysia’s Antimicrobial Resistance Integrated Surveillance program. The test using the disk diffusion method for *E. coli* was conducted according to the 2018 Clinical and Laboratory Standards Institute (CLSI) guidelines [43], whereas that for *Vibrio* was performed according to the CLSI guidelines of 2016 and 2020 [44,45]. The determination of the minimum inhibitory concentration (MIC) of colistin was carried out according to the CLSI guidelines 2012 and 2018 [43,46]. The isolates demonstrating resistance to more than three groups of antibiotics were categorized as MDR [47]. We identified MDR in *n* = 110 *E. coli* and *n* = 35 *V. parahaemolyticus* isolates. All the work was carried out at the Veterinary Public Health Laboratory, Faculty of Veterinary Medicine, Universiti Putra Malaysia.

### 2.2. Data Collection

A structured questionnaire was adapted from previous publications using the guidance provided by the OIE [41,48]. The questionnaire comprised open- and closed-ended questions about demography, farm management and production practices and was used to retrieve information on the putative risk factors for the carriage of MDR *E. coli* and *V. parahaemolyticus* in the aquaculture settings. The questionnaire was pre-tested, and content validity was conducted by the experts in the field of veterinary public health, epidemiology, aquatic animal health and AMR. The questions were refined based on the suggestions and observations made to improve the content validity. Prior to data collection, the farmers were assured of confidentiality and briefed about the research objectives, followed by the signing of an informed consent document. Face-to-face administration of the questionnaire was performed by the researchers.

The following variables were investigated in this study:

**Type of production system:** the production systems used for cultivating tilapia and Asian seabass are pond (earthen pond) and cage-culture systems. Pond culture is the rearing of fish in the natural inland aquaculture or artificial basins. Ponds can be categorized based on pond constructions; for instance, an earthen pond is constructed entirely from the soil. Cage culture involves cultivating the fish within fixed or floating net enclosures braced by a framework constructed from natural materials such as wood or modern polyethylene, and set in sheltered, shallow portions of lakes, rivers, estuaries or oceans [49,50].

**Water exchange:** describes a system that allows continuous replacement of fresh surface water.

**History of diseases:** a fish farm is considered positive if the fish were exposed to disease during their growing stage.

**Manure fertilization:** refers to the utilization of animal manure to increase the production of natural food organisms in the grow-out ponds [51]. This application is traditionally practiced in a pond-system aquaculture setting [52,53].

**History of antibiotic application**: refers to the utilization of any form of antibiotics during the growing stage of the fish.

**Human activities:** this is indicated by the presence of residential areas and commercial developments such as markets, as well as recreational fishing spots, within a 1 km radius of the aquaculture farm.

**Livestock near the farm:** refers to sightings of cattle, goat or poultry farms within a 1 km radius from the aquaculture farm.

### 2.3. Data Analysis

The antimicrobial sensitivity and multidrug resistance test data of *E. coli* and *V. parahaemolyticus* isolates recovered from tilapia or Asian seabass were analyzed separately using WHONET 5.6 [54,55]. A multilevel binary logistic regression was employed because our data was hierarchical in nature, with bacterial isolates nested within farms [56,57,58]. The analysis was performed following the steps suggested by and Sommet and Morselli [57] and Crowson [56].

The chi-square test was used for the univariable exploratory analysis to identify a priori risk factors associated with the outcome variable. Variables for inclusion in the full model were selected based on the univariable analysis (*p* < 0.05) and variables that were suggestive of important effects. The model building used a forward stepwise selection process [59,60]. All the statistical analyses were performed using the IBM SPSS Statistics version 26 (IBM, Armonk, NY, USA: IBM Corp.) at a significance level of α = 0.05.

## 3. Results

### 3.1. MDR Escherichia coli

#### 3.1.1. Descriptive Statistics

Thirty-two fish farms cultivating tilapia (*n* = 19) and Asian seabass (*n* = 13) were included in this study, which consisted of earthen pond (*n* = 20; 62.5%) and cage-culture (*n* = 12; 37.5%) systems. Most of the tilapia sampled were raised in an earthen pond (94.7%), while most of the seabass (76.9%) were raised in a cage-culture system. Regular water exchange occurs in 30 farms (93.7%). The cage-culture farms do not practice the application of animal manure, while three of the pond system farms (9.4%) administer manure-based fertilizers to increase the production of the pond’s natural food organisms. Two of the farms (6.3%) (earthen-pond farms) reported a history of antibiotics use for treatment of fish disease, and the remaining 30 farms (93.7%) reported not using any antibiotics. Five farms (15.6%) reported a history of disease infection, while the remaining 27 farmers (84.4%) reported no disease on their farms. Pets such as cats and dogs can be seen in 16 farms (50%), while 16 farms (50%) reported no pets. Anthropogenic activities were observed nearby in the study area; 18 of the farms (56.3%) were located near human activity, and 8 farms (25%) were located near livestock farms. Human activities observed within 1 km of the fish farms were residential areas, commercial development (e.g., markets), recreational fishing spots area and oil palm agriculture. Livestock farms such as poultry, goat and cattle were observed within 1 km of the fish farms in the study area. All the variables in this study presented in Table 1.

#### 3.1.2. Univariable Factor Associated with MDR *E. coli*

In the univariable analysis, the factors significantly associated with the occurrence of MDR among *E. coli* isolated from aquaculture production systems were the type of production system (χ^2^ = 8.328; *p* = 0.004), use of manure in the pond (χ^2^ = 16.213; *p* = 0.000), presence of pets in the farm (χ^2^ = 6.079; *p* = 0.014) and livestock observed in nearby areas (χ^2^ = 6.967, *p* = 0.008) (Table 1).

#### 3.1.3. Multivariable Factor Associated with MDR *E. coli*

The full model of multilevel binary logistic regression analysis revealed that manure application (OR = 4.5; 95% CI = 1.21–16.57; *p* = 0.025) was significantly associated with an increased chance of MDR *E. coli* in the aquaculture system (Table 2). *Escherichia coli* isolated from the aquaculture farms using manure fertilization in the pond were approximately 4.5 times more likely to acquire MDR as compared with those from farms that did not use manure fertilization.

### 3.2. MDR Vibrio parahaemolyticus

#### 3.2.1. Descriptive Statistics

Thirteen Asian seabass farms consisting of three earthen ponds (23.1%) and ten cage-culture (76.9%) production systems participated in this study. Only one of the farms (7.7%) reported a history of antibiotic administration, and the remaining twelve farms (92%) reported no antibiotic usage. Four farms (30.8%) had a previous history of disease outbreaks, while the remaining nine farms (69.2%) had none. Pets such as cats and dogs can be observed in eight farms (61.5%), while five farms (38.5%) reported no pets. Anthropogenic activities were observed nearby; four of the farms (30.8%) were located near areas of human activities, and one farm (7.6%) was located near a livestock farm. All the variables in this study presented in Table 3.

#### 3.2.2. Univariable Factors Associated with MDR *V. parahaemolyticus*

In the univariable analysis, the factors significantly associated with the occurrence of MDR among *V. parahaemolyticus* isolates recovered from the Asian seabass production system were the type of production system (χ^2^ = 58.884; *p* = 0.000), antibiotic usage (χ^2^ = 9.640; *p* = 0.002), human activity nearby (χ^2^ = 51.857; *p* = 0.000), presence of pets in the farm (χ^2^ = 3.866; *p* = 0.049) and livestock nearby (χ^2^ = 9.640; *p* = 0.002) (Table 3).

#### 3.2.3. Multivariable Factor Associated with MDR *V. parahaemolyticus*

The full model of the multilevel binary logistic regression analysis revealed that the earthen pond system (OR = 8.2; 95% CI = 1.47–45.2; *p* = 0.017) was significantly associated with eight times increased odds of MDR *V. parahaemolyticus* in Asian seabass farms (Table 4). In addition, although not statistically significant, human activity adjacent to the farm elevated the odds of MDR (OR = 4.6; 95% CI = 0.75–27.98; *p* = 0.099) amongst the isolates.

## 4. Discussion

The aquaculture production system may serve as an important receptacle of MDR bacteria and ARGs, enabling their broad transference throughout the aquatic ecosystems, thus endangering food safety and security [61,62,63]. The literature suggests that activities in fish farming may drive AMR and MDR development [64,65,66,67]. Therefore, this study explores these activities.

### 4.1. Escherichia coli

We found that using animal manure as a pond fertilizer significantly increased MDR in the *E. coli* recovered from our aquaculture setting. This finding is supported by other similar studies such as a study from Thailand that reported a higher level of AMR in bacteria recovered from the fish gut raised in integrated broiler-fish farms that dispensed manure into the pond [68] and complemented by recent studies conducted in Tanzania and Bangladesh that found an accumulation of ARGs in the fish ponds that use domestic farm and poultry wastes from animal husbandry [69]. More recently, a report by Thornber et al. [67] identified that using human and livestock/poultry wastes to fertilize ponds in the traditional integrated farming practices is associated with elevated AMR risk in shrimp farming. Furthermore, a study by Adeyemi et al. [64] demonstrated that adding poultry droppings to the fish pond promotes development of MDR in gram-negative bacteria in the water and fishes.

Using manure in aquaculture ponds aims to establish a condition that can expedite a surge in the stocks of natural foods and boost a prebiotic effect that positively stimulates alterations of the pond microbiota [70]. However, Lima et al. [71] and Fatoba et al. [72] found that using manure may increase the risk of indicator bacterial pathogens developing MDR because antibiotics are widely used in food animal production. Farmers in Asian countries administer higher amounts of antibiotics in food animals as compared to those in other parts of the world. Van Boeckel et al. [73] projected that food animals in the Southeast Asian countries will have the highest increase in antibiotic use by 2030. About 52–276 mg of antibiotics per kilogram of live chicken and 46 mg of antimicrobial agents per kg of live pig were found in Southeast Asian countries; furthermore, 286.6 mg and 77.4 mg of in-feed antimicrobials were utilized to grow 1 kg of live pig and poultry, respectively [74], compared to 26.1–147.3 mg/PCU (population correction unit) in pigs and 107.3 mg/PCU in poultry in European countries [73]. Therefore, it is not surprising that livestock manure contains antibiotics in its parental form [75]. For aquaculture, the interconnectivity of aquatic ecosystems augments the AMR and MDR further, as locally generated resistance can easily be dispersed to other global settings [66].

Another major concern with raising fish aimed for human consumption in manure-fertilized ponds is the presence of partially degraded or undegraded antibiotics. The bioaccumulation of these antibiotics generates continuous selective pressure on the aquatic microbiota [70]. Aquatic microbes harbor numerous mobile genetic elements, such as plasmids and transposons, that can recombine and mobilize, thus enhancing the emergence of AMR and MDR through horizontal transfer of ARGs among bacterial pathogens [15,62,76]. These elements can travel to the wider environment and human habitat [61,77,78]. Fertilization of fish ponds using manure should, therefore, be removed to minimize the risk of bacterial pathogens developing and acquiring MDR. At the very least, if manure is to be used, it should be pretreated by sterilization and fermentation [79,80].

### 4.2. Vibrio parahaemolyticus

We found that the earthen pond production system significantly increases the likelihood of MDR *V. parahaemolyticus*; isolates recovered from the earthen pond of Asian seabass culture were eight times more likely to demonstrate MDR compared to those recovered from the cage system. Pond water quality is highly vulnerable to environmental conditions, including pollutants [81,82]. When ponds are not or rarely emptied throughout the production cycle, antimicrobials and antimicrobial residues can rapidly accumulate in the sediments [68]. In addition, long intervals between pond sediment removal results in the rapid accumulation of antibiotics [83], exerting continuous selective pressure that stimulates the development and selection of antimicrobial-resistant bacteria in the fish ponds [68,84,85]. On the other hand, cage culture is characterized by fixed or floating net enclosures braced by a framework constructed with natural materials, such as wood or modern polyethylene, and set in estuaries, rivers or oceans. The cages are susceptible to waves, seasonal change, temperature and other environmental factors [86,87]. The lower MDR level found in the present study among isolates recovered from the cage culture production may partly be attributed to the effect of antibiotic residues or resistance elements that are diluted and dispersed by waves and degradation [88].

*V. parahaemolyticus* obtained from farms near areas where human activities take place, such as residential areas and recreation sites such as fishing spots, faced an increased risk of MDR. This finding is supported by several studies, such as by Zhang et al. and Borella et al. [89,90], that linked resistance genes in the marine aquaculture to terrestrial and anthropogenic sources. Pruden et al. [91] found that upstream human activities facilitate the dissemination of riverine ARGs. Ghaderpour et al. [92] also documented high resistance levels and MDR *E. coli* in the aquaculture environment located upstream of the Selinsing and Sangga Besar rivers in Malaysia, which receive anthropogenic runoffs. In addition, a recent study conducted in the river estuary in Johor, Malaysia, found that the sampling site where residential areas and sewage treatment plants were clustered together had frequent microbial abundance and MDR profile [93]. Zheng et al. [94] reported that estuarine and coastal environments polluted by antibiotics and antibiotic-resistant genes frequently receive a series of contaminants such as riverine runoff, wastewater treatment plants, sewage discharge and aquaculture.

### 4.3. Limitations

Our study should be interpreted with caution because of several limitations. Misclassification bias could have occurred because of poorly documented antibiotic use at the farms. Many farmers do not have proper documentation and records of diseases and treatments administered on their farms. In addition, we were unable to capture differences as a result of changes in the water characteristics across time and season since sampling was done just once for each farm.

## 5. Conclusions

This study highlights that the use of livestock manure for pond enrichment is an important driver in the development of MDR *E. coli* in aquaculture systems. Hence, it is recommended to reduce or avoid manure application altogether, or if this method is still preferred, then the manure must be pretreated before adding it to the pond. Furthermore, earthen pond systems and anthropogenic activities near the farms elevate the likelihood of MDR *V. parahaemolyticus*. This study’s findings inform aquaculture interventions that are likely to effectively mitigate MDR emergence amongst bacterial pathogens of public health and animal health importance. It may also reduce the risk of the local aquaculture setting becoming the mixing vessel of various resistant pathogens, as well as resistant genes that could be rerouted to the global aquatic ecosystems.

## Figures and Tables

**Table 1 microorganisms-10-01520-t001:** Univariable analysis for risk factors associated with the growth of MDR *E. coli* in the aquaculture production systems (*n* = 32) on the west coast of peninsular Malaysia.

Variables	Frequency	Positive (%)	Chi-Square (χ^2^)	*p*-Value
Human activity near farm				
Yes	161	77 (47.8)	2.460	0.117
No	88	33 (37.5)		
Livestock near farm				
Yes	80	45 (56.3)	6.967	0.008 *
No	169	65 (38.5)		
Type of fish				
Tilapia	202	94(46.5)	2.413	0.120
Asian seabass	47	16 (34)		
Type of production system				
Earthen pond	186	92 (49.5)	8.328	0.004 *
Cage	63	18 (28.6)		
Water exchange				
Yes	236	102 (43.2)	1.677	0.195
No	13	8 (61.5)		
Manure application				
Yes	36	27 (75)	16.213	0.000 *
No	213	83 (39)		
Diseases history				
Yes	37	14 (37.8)	0.708	0.400
No	212	96 (45.3)		
Antibiotic application				
Yes	16	10 (62.5)	2.328	0.127
No	233	100 (42.9)		
Presence of pets in farm (dogs, cats)				
Yes	139	71 (51.1)	6.079	0.014 *
No	110	39 (35.5)		

* Statistically significant, *p* < 0.05.

**Table 2 microorganisms-10-01520-t002:** Multilevel binary logistic regression of risk factors associated with the occurrence of MDR *E. coli* in the aquaculture production system (*n* = 32) on the west coast of peninsular Malaysia.

Variables	Coefficient	SE	t	*p*-Value	OR	95% Confidence Interval
Intercept	−0.424	0.2298	−1.846	0.066	0.65	0.42–1.03
Manure application (Yes)	1.500	0.6636	2.261	0.025	4.48	1.21–16.57
Water exchange (Yes)	0.903	0.8212	1.099	0.273	2.47	0.49–12.43

SE, standard error; OR, odds ratio.

**Table 3 microorganisms-10-01520-t003:** Univariable analysis for risk factors associated with the occurrence of MDR *V. parahaemolyticus* in the Asian seabass production system (*n* = 13) on the west coast of peninsular Malaysia.

Variables	Frequency	Positive (%)	Chi-Square (χ^2^)	*p*-Value
Human activity near farm				
Yes	63	32(50.8)	51.857	0.000 *
No	99	3(3.0)		
Livestock near farm				
Yes	21	10(47.6)	9.640	0.002 *
No	141	25(17.7)		
Type of production system				
Earthen pond	52	30(57.7)	58.884	0.000 *
Cage	110	5(4.5)		
Diseases history				
Yes	62	13(21)	0.024	0.877
No	100	22(22)		
Antibiotic application				
Yes	21	10(47.6)	9.640	0.002 *
No	141	25(17.7)		
Presence of pets in farm (dogs, cats)				
Yes	69	20(29)	3.866	0.049 *
No	93	15(16.1)		

* Statistically significant, *p* < 0.05.

**Table 4 microorganisms-10-01520-t004:** Multilevel binary logistic regression of risk factors associated with the occurrence of MDR *V. parahaemolyticus* in the Asian seabass production system (*n* = 13) on the west coast of peninsular Malaysia.

Variables	Coefficient	SE	t	*p*-Value	OR	95% Confidence Interval
Intercept	−3.025	0.4783	−6.325	0.000	0.049	0.019–0.125
Type of production system (earthen pond)	2.099	0.867	2.421	0.017	8.16	1.47–45.23
Human activity near farm (Yes)	1.522	0.9163	1.661	0.099	4.58	0.75–27.98
Livestock near farm (Yes)	−0.691	0.5760	−1.200	0.232	0.50	0.16–1.56

SE, standard error; OR, odds ratio.

## Data Availability

The data presented in this study are available in the manuscript.

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
