# Peer review of "On-Farm Practices Associated with Multi-Drug-Resistant Escherichia coli and Vibrio parahaemolyticus Derived from Cultured Fish"

_microorganisms, 2022, doi:10.3390/microorganisms10081520_

Round 1

Reviewer 1 Report

This manuscript “On-farm Practices Associated with Multi-drug Resistant Escherichia coli and Vibrio parahaemolyticus Derived from cultured Fish” prepared by Dewi et al. presented the targeted interventions associated with the risk of AMR bacteria. The topic of the manuscript is of great interest to bacterial antibiotic resistance development and fish farming. The mathematic models in analyzing antibiotic resistance are appreciated, and revisions clarifying the importance of the findings will be needed.

Major:

  1. In the introduction, please introduce why the antibiotics were chosen to be tested and what are the bacterial antibiotic resistance mechanisms associated with them? Have people shown gene transference targeting that antibiotic resistance?
  2. Line 87: I understand this study is the follow-up of a previous study. But the origins of bacteria are of great importance to the analysis in this manuscript. Including the bacteria isolation in the method section will be greatly apperciated. 
  3. How did you define the bacteria isolates to be unique from the other ones? How many of them are from the same fish? Which part of the fish did you isolate the bacteria from? How many fish do you use for each farm? Are they evenly distributed from all farms? 
  4. For the antibiotic-resistant assay, how many times have you replicated your experiments? And you can consider using susceptibility instead of resistance as a more accurate way in describing the phenotype.
  5. Line 103: what do you mean by "three groups"? Three different kinds of antibiotics? Biofilm formation has been shown as an important factor in antibiotic resistance for both E. coli and V. para, have you considered this as a potential factor?
  6. Are all assumptions met for the multivariable logistic regression? I am a bit concerned about the sample size in this manuscript. Is sample size efficient in accurately estimating small fixed effects (see this paper for details: https://www.tandfonline.com/doi/full/10.1080/00220973.2015.1027805). I would suggest a Monte Carlo study as compensation for the insufficient sample size.

Minor:

1. 3.1.1 add a reference to table 1.

2. Table 1-3, add significant P-value standard: what do you think is statistically significant?

3. Line 108: Data Collection, please check the spelling and grammar.

4. Please be sure all bacteria names are italicized in the manuscript.

Author Response

Dear Reviewer,

Thank you for the opportunity to revise our manuscript with entitle “On-farm Practices Associated with Multi-drug Resistant Escherichia coli and Vibrio parahaemolyticus Derived from Cultured Fish” (ID: Microorganisms- 1813686). We appreciate your comments and trust that the comments the quality of our paper.  Below is the point by point response from your comments:

Point 1 

Comments and Suggestions for Authors

This manuscript “On-farm Practices Associated with Multi-drug Resistant Escherichia coli and Vibrio parahaemolyticus Derived from cultured Fish” prepared by Dewi et al. presented the targeted interventions associated with the risk of AMR bacteria. The topic of the manuscript is of great interest to bacterial antibiotic resistance development and fish farming. The mathematic models in analyzing antibiotic resistance are appreciated, and revisions clarifying the importance of the findings will be needed.

Authors’ response: Thank you for the reviewer’s comments and recognition of the study

Major Comments

Point 2

In the introduction, please introduce why the antibiotics were chosen to be tested and what are the bacterial antibiotic resistance mechanisms associated with them? Have people shown gene transference targeting that antibiotic resistance?

Authors’ response: We appreciate your comment. We have discussed the reason for antibiotic selection in the materials and methods of the manuscript (lines 102-104). Our reasons for selecting these antibiotics were due to the WHO and OIE recommendations for antimicrobial usage in humans and livestock, as well as alignment with the Malaysia's Antimicrobial Resistance Integrated Surveillance program. We agree that it could be of interest to include the resistance mechanisms and gene transference targeting these antibiotic resistances in the Introduction, however in keeping the manuscript concise, we feel that this aspect is less relevant since our objectives and scope of work did not cover the analyses of resistance genes.  

Point 3

Line 87: I understand this study is the follow-up of a previous study. But the origins of bacteria are of great importance to the analysis in this manuscript. Including the bacteria isolation in the method section will be greatly appreciated. 

Authors’ response: Thank you for the suggestion. Accordingly, we have revised the method section (lines 89-96).

Point 4

  1. How did you define the bacteria isolates to be unique from the other ones?
  2. How many of them are from the same fish?
  3. Which part of the fish did you isolate the bacteria from?
  4. How many fish do you use for each farm?
  5. Are they evenly distributed from all farms? 

Authors’ response: Thank you for the comments. We hope that we understood your questions correctly and we addressed them below:

  1. We used standard methods to culture and identify coli and Vibrio from the samples (fish gut and pond water). In addition, since each isolates came from different fish we assumed that the isolate is unique. Molecular analysis by using polymerase chain reaction (PCR) methods was applied to define the species of Vibrio. We further did the typing of V. parahaemolyticus by using the multi-locus sequence technique (MLST) for epidemiological molecular analysis (MLST is not included in this manuscript).
  2. Based on our database, about 19% coli and V. parahaemolyticus originated from the same fish.
  3. We isolated the bacteria from the intestine of fish.
  4. We obtained the fish 20-30 fishes each farm.
  5. We cannot assume that the fishes we took are evenly distributed because we sample fish at market age therefore depending on the farm, the fish may be limited at certain area of the farm. However, within a sampled pond, we assume that the fishes we sampled were well distributed given that they swam freely in the pond.

Point 5

For the antibiotic-resistant assay, how many times have you replicated your experiments? And you can consider using susceptibility instead of resistance as a more accurate way in describing the phenotype.

Authors’ response: Thank you for the comment. We ran two replicates of antibiotic susceptibility test (AST) for E. coli and V. parahaemolyticus isolates and we used the reference quality control strains  E. coli ATCC 25922 (CLSI, 2018) in all our experiments to standardize the quality control of antibiotic-susceptibility assays.

Thank you for the suggestion regarding the phenotype of resistance however we prefer to use resistance since our focus is MDR.

Point 6

Line 103: what do you mean by "three groups"? Three different kinds of antibiotics? Biofilm formation has been shown as an important factor in antibiotic resistance for both E. coli and V. parahaemolyticus, have you considered this as a potential factor?

Authors’ response: Thank you. “Three groups” in line 103 of manuscript refer to three antimicrobial classes. Multi-drug resistance (MDR) is defined as resistant to three or more antimicrobial classes (Magiorakos et al., 2012).

We did not include biofilm formation in the aquatic system as a potential factor in antibiotic resistance for E. coli and V. parahaemolyticus in this project as we did not analyse for biofilm. Indeed, we agree that biofilm formation in the aquatic system especially in aquaculture may play a role as one of the drivers of antibiotic resistance in aquaculture since its structure allows for the genetic exchange among high population densities of microbes and proximity of cells in the biofilms (Balcazar.et al., 2015). We will consider your valuable input in our future research project.

Point 7

Are all assumptions met for the multivariable logistic regression? I am a bit concerned about the sample size in this manuscript. Is sample size efficient in accurately estimating small fixed effects (see this paper for details: https://www.tandfonline.com/doi/full/10.1080/00220973.2015.1027805). I would suggest a Monte Carlo study as compensation for the insufficient sample size.

Authors’ response: Thank you for the suggestion. We have checked the assumption for the multivariable logistic regression such as:

There is no multicollinearity in the data: analysis of collinearity statistics showed this assumption has been fulfilled, as VIF scores were below 10  (with ranges between 1-3), and tolerance scores above 0.2 both for E. coli and V parahaemolyticus.

The values of the residuals are independent; the Durbin-Watson statistic showed that this assumption had been met, as the obtained value was close to 2 (Durbin-Watson = 1.63 (E. coli) and 1.54 (V. parahaemolyticus)).

The variance of the residuals is constant. Our plot of standardized residuals vs standardized predicted values showed no obvious signs of funneling, suggesting the assumption of homoscedasticity has been met in our data.

There are no influential cases biasing in the model. Cook’s Distance values were all under 1, suggesting individual cases were not unduly influencing the model.

In addition, Intra-classes correlation (ICC) is a primary method for researchers to determine whether multi-level model is necessary. The null model showed significant (p < 0.05) variation in the MDR among E. coli isolate among farms; in particular, ICC indicated that 20% of the chance of MDR is explained by between-farm differences. As for V. parahaemolyticus isolates among farms; in particular, the ICC indicated that 55% of the chance of MDR is explained by between-farm differences, suggesting the chance of V. parahaemolyticus acquired the MDR varied among fish farm.

We appreciate your suggestion regarding the sample size. However, we assumed that E. coli sample size is sufficient enough to accurately estimate small fixed effects since we have 32 groups of level-2 units (McNeish et al., 2016). Moreover, the fixed effects estimates were unbiased even with 30 groups (Ali et al., 2019). Unfortunately, V. parahaemolyticus sample size has less than 30 groups of level-2 units. However, Maas & Hox (2004) carried out simulation study regarding sample size issues in multilevel models and they determined that for fixed effects 10 groups are sufficient. In addition, since all the data met the assumption to be analyzed by using multivariable binary logistic, we assumed that our data could be presented as if with the cautions of the sample size violation.

Minor comments

Point 8

 3.1.1 add a reference to table 1.

Authors’ response:  Thank you for the suggestion. We accordingly have revised the paper (lines 183 and 227)

Point 9

Table 1-3, add significant P-value standard: what do you think is statistically significant?

Authors’ response: Thank you for the suggestion. We have revised the table 1-3 of paper with add the P-value standard at a significance level of α = 0.05 (line 203, 247)

Point 10

Line 108: Data Collection, please check the spelling and grammar.

Authors’ response: Thank you. Accordingly, we have revised the spelling and grammar (line 114).

Point 11

Please be sure all bacteria names are italicized in the manuscript.

Authors’ response:  Thank you for the suggestion. We have checked all the bacteria names are italicized.

Sources:

Ali A, Ali S, Khan SA, Khan DM, Abbas K, Khalil A, et al. (2019) Sample size issues in multilevel logistic regression models. PLoS ONE 14 (11): e0225427. https://doi.org/10.1371/journal.pone.0225427

Balcázar JL, Subirats J and Borrego CM (2015) The role of biofilms as environmental reservoirs of antibiotic resistance. Front. Microbiol. 6:1216. doi: 10.3389/fmicb.2015.01216

CLSI Standard VET01; Performance Standards for Antimicrobial Disk and Dilution Susceptibility Test for Bacteria Isolated
from Animals. 5th ed. Clinical and Laboratory Standards Institute: Wayne, PA, USA, 2018; 19–50.

Magiorakos, A.P., Srinivasan, A., Carey, R.B., Carmeli, Y., Falagas, M.E., Giske, C.G., Harbarth, S., Hindler, J.F., Kahlmeter, G., Olsson-Liljequist, B., Paterson, D.L., Rice, L.B., Stelling, J., Struelens, M.J., Vatopoulos, A., Weber, J.T., Monnet, D.L. Multidrug-resistant, extensively drug-resistant and pandrug-resistant bacteria: An international expert proposal for interim standard definitions for acquired resistance. Clin. Microbiol. Infect. 2012, 18, 268–281. https://doi.org/10.1111/j.1469-0691.2011.03570.x

Maas C.J. and Hox J.J., “Robustness issues in multilevel regression analysis”, Statistica Neerlandica. 2004; 58(2), 127–37

McNeish, D.M., Stapleton, L.M. The Effect of Small Sample Size on Two-Level Model Estimates: A Review and Illustration. Educ Psychol Rev 28, 295–314 (2016). https://doi.org/10.1007/s10648-014-9287-x

Sommet, N., Morselli, D., 2017. Keep calm and learn multilevel logistic modeling: A simplified three-step procedure using stata, R, Mplus, and SPSS. Int. Rev. Soc. Psychol. 30, 203–218. https://doi.org/10.5334/irsp.90

Yang, J., Zhao, Z., Li, Y., Krewski, D., Wen, S.W., 2009. A multi-level analysis of risk factors for Schistosoma japonicum infection in China. Int. J. Infect. Dis. 13. https://doi.org/10.1016/j.ijid.2009.02.005

Reviewer 2 Report

To avoid the emergence and diffusion of AR and MDR bacteria from aquaculture centers appear very important in that it is know that these centers are contributing more and more to feeding the human population. The interest of this study is that it is aimed to identify the factors/activities promoting the emergence of resistant bacteria. It seems clear that any condition allowing antibiotics to enter the water and sediments where fishes are cultured is a factor of risk for the selection of AR bacteria: not only the direct  somministration of antibiotics is a risk but also the introduction of antibiotics present in manure and faeces and other materials from farm animals and pets.

The study is well designed and presented but I cannot well understand why E.coli and V. parahaemolyticus have been analyzed separately and considering different at risk conditions. Please explain that decision. Why other resistant bacteria have not been considered ? I think authors should make a general consideration referred to the two, and any other AR bacteria, species in that it doesnt seem that different bacterial species might be influenced by different factors.

In the text it should be appropiate to associate the reference sentences to the name of the  author publishing them and not only to the reference number:

not ''[86] also documented high resistance levels and MDR E. coli in the aquaculture environment''

but Gatherpur et al. [86]  also documented high resistance levels and MDR E. coli in the aquaculture environment

Author Response

Dear Reviewer,

Thank you for the opportunity to revise our manuscript with entitle “On-farm Practices Associated with Multi-drug Resistant Escherichia coli and Vibrio parahaemolyticus Derived from Cultured Fish” (ID: Microorganisms- 1813686). Here is authors’ response

Comments and Suggestions for Authors

Point 1

To avoid the emergence and diffusion of AR and MDR bacteria from aquaculture centers appear very important in that it is know that these centers are contributing more and more to feeding the human population. The interest of this study is that it is aimed to identify the factors/activities promoting the emergence of resistant bacteria. It seems clear that any condition allowing antibiotics to enter the water and sediments where fishes are cultured is a factor of risk for the selection of AR bacteria: not only the direct administration of antibiotics is a risk but also the introduction of antibiotics present in manure and faeces and other materials from farm animals and pets.

The study is well designed and presented but I cannot well understand why E. coli and V. parahaemolyticus have been analyzed separately and considering different at risk conditions. Please explain that decision.

Authors’ response: Thank you for the reviewer’s comments and recognition of the study. Our reason for this separation is due to the different nature of the microorganisms. E. coli is known as sentinel bacteria in the aquatic system which generally originate from the terrestrial warm-blooded animal. Meanwhile, V. parahaemolyticus is an aquatic bacteria that inhabit the water system. Therefore, these two organisms exist through different ways in aquaculture. E. coli are found both in freshwater and marine aquaculture, meanwhile, V. parahaemolyticus is a strictly halophilic bacteria that is only found in marine fish. In this study, E. coli is obtained from tilapia and Asian seabass while V. parahaemolyticus is only found in Asian seabass. Therefore, different risks condition will impact these two species differently so analysis has to also be done separately to allow these risks to be captured more accurately.

Point 2

Why other resistant bacteria have not been considered? I think authors should make a general consideration referred to the two, and any other AR bacteria, species in that it doesn’t seem that different bacterial species might be influenced by different factors.

Authors’ response: We hope we understood your question correctly. We focus on these two bacteria based on the criteria for microorganisms chosen in guideline of surveillance and monitoring of antibiotic usage and AMR in aquatic animals which is published by the OIE (OIE, 2015). E. coli is an important AMR indicator bacteria and is an efficient resistance gene material donor. At the same time, it is also an important pathogen for most species of animals. Across AMR surveillance worldwide, E. coli is consistently monitored over time.  While Vibrio is a very important aquatic organism and foodborne pathogen and is the only aquatic organism consistently monitored in AMR surveillance. In Malaysia, the two species are targeted for surveillance  in the Malaysian Strategic Action Plan for AMR 2017-2021 (MyAP–AMR, Ministry of Health Malaysia (2017). Furthermore, consistent monitoring of the two species will allow various work on AMR to be compared across labs and geography.

Point 3

In the text it should be appropriate to associate the reference sentences to the name of the author publishing them and not only to the reference number:

not ''[86] also documented high resistance levels and MDR E. coli in the aquaculture environment''

but Gatherpur et al. [86] also documented high resistance levels and MDR E. coli in the aquaculture environment

Authors’ response: Thank you for the suggestion, according to the comments we have revised the references sentences (Line 43, 93, 96, 157, 274, 277, 282, 286, 326, 328, 329, 334).

Sources:

Ministry of Health Malaysia. (2017). Malaysian Action Plan on Antimicrobial Resistance (MyAP-AMR) 2017-2021. In Ministry of Health Malaysia.

OIE. (2015). OIE Standards , Guidelines and Resolution on antimicrobial resistance and the use of antimicrobial agents. https://web.oie.int/delegateweb/eng/ebook/AF-book-AMR-NG_FULL.pdf?WAHISPHPSESSID=03152ead00d06990fa9066b7b71fcabc

Round 2

Reviewer 1 Report

Thank you for answering all the questions!